# Evaluation of the Anticancer and Probiotic Potential of Autochthonous (Wild) *Lacticaseibacillus paracasei* Strains from New Ecological Niches as a Possible Additive for Functional Dairy Foods

**DOI:** 10.3390/foods12010185

**Published:** 2023-01-01

**Authors:** Ekaterina Vachkova, Valeria Petrova, Natalia Grigorova, Zhenya Ivanova, Georgi Beev

**Affiliations:** 1Department of Pharmacology, Animal Physiology and Physiological Chemistry, Faculty of Veterinary Medicine, Trakia University, 6000 Stara Zagora, Bulgaria; 2Department of Biological Sciences, Faculty of Agriculture, Trakia University, 6000 Stara Zagora, Bulgaria

**Keywords:** symbiotic microbial products, non-differentiated and differentiated HT29 cell line, alkaline phosphatase activity, IAP gene expression, Bax/Bcl-2 ratio

## Abstract

Probiotics such as *Lactobacillus* spp. could modulate the intestinal microbiota composition, supporting gastrointestinal tract barrier function and benefiting human health. To evaluate the anticancer and probiotic properties of potentially active autochthonous *Lacticaseibacillus paracasei* strains on proliferating and differentiated enterocytes, human colon adenocarcinoma cell line HT29 was used as a model. The lactic acid bacteria (LAB) were isolated from new ecological niches—mountain anthills populated by redwood ants (*Formica rufa* L.). Human colorectal adenocarcinoma cells (HT29, ATCC, HTB-38™) were treated for twenty-four hours with supernatants (SNs) derived from four strains of *Lacticaseibacillus paracasei*: P4, C8, C15 and M2.1. An MTT assay, alkaline phosphatase activity, IAP, Bax and Bcl-2 gene expression analysis (RT-qPCR) and the Bax/Bcl-2 ratio were evaluated. The MTT assay revealed that the observed effects varied among groups. However, 10% neutralized supernatants from P4, C8, C15 and M2.1 strains did not show cytotoxic effects. In contrast to non-differentiated cells, a significant (*p* < 0.001) rise in ALP activity in all treatments, with an average of 18%, was established in differentiated cells. The IAP expression was remarkably downregulated in the differentiated M2.1 group (*p* < 0.05) and upregulated in the non-differentiated P4 (*p* < 0.05) and M2.1 (*p* < 0.05) groups. The Bax/Bcl-2 quantity expression ratio in P4 was significantly (*p* < 0.05) upregulated in proliferating cancer cells, but in P4- and M2.1-differentiated cells these values were downregulated (*p* < 0.05). The obtained results indicate that the isolated *L. paracasei* strains possess anticancer and probiotic properties and could be used as additives for functional dairy foods and thus benefit human health.

## 1. Introduction

Recent trends for the enrichment of the diet with factors capable of preventing disorders in the organism’s immune status and reducing the risk of diseases could become one of the major approaches to preventive treatment in humans [1,2]. The advent of functional foods or nutraceuticals objectively emphasizes alternatives to limit the use of drugs that promote the regular consumption of fermented foods. When probiotics are consumed frequently and in adequate amounts, their usage could benefit human health. Probiotic and lactic acid bacteria (LAB), in particular, have been recommended to fulfill the role of nutraceuticals, as they have no side effects for human health and also can stimulate the immune system and thus potentiate its resistance against numerous disease conditions [3]. Including LAB with high probiotic potential in starter cultures for dairy products that have a significant proportion in the human diet could be used to prevent and treat many diseases [4,5]. This requires the identification of different natural sources from which to isolate suitable LAB with valuable probiotic, functional and technological characteristics to produce high-quality dietary and healthy dairy products [6,7]. With this in mind, we have attempted to isolate a new wild LAB strain following some ancient and forgotten practices of a small ethnic group of shepherds (Karakachan) living in the Bulgarian Balkan Mountains. According to narratives, they used substrates from alpine anthills to produce traditional Bulgarian-style yogurt from sheep milk.

*Lactobacillus* species are non-pathogenic lactic-acid-producing bacteria (LAB) that display probiotic properties and are technologically suitable for industrial processes [8]. *Lacticaseibacillus paracasei* (*L. paracasei*) is a widespread species found in yogurt and milk [9], koumiss [10,11], cheese [12,13,14,15], fruits, flower inflorescences [8], honey [16], sourdough bread [17], fish intestinal tracts [18], etc.

The health benefits of probiotics include some disease conditions such as lactose intolerance, diabetes, obesity, acute diarrheal disease, inflammatory bowel diseases, irritable bowel syndrome, cancer, cardiovascular diseases, urogenital infections, allergies, the gut–brain axis, the antiviral activity of lactic acid bacteria, etc. [3]. Probiotics increase and maintain the gut mucosal barrier by stimulating goblet cells [19]. The interaction between *Lactobacillus* spp. and intestinal epithelial cells can cause the differentiation of immune cells and regulate the gut’s barrier function [19]. In the food–gut complex ecosystem, dairy LAB can induce a network of signals mediated by the whole bacteria or their components (peptidoglycans, exopolysaccharide extracts, cellular extract) [20], and even death; they can also exhibit probiotic [21,22] and anticancer activity [23,24,25]. Another functional axis is the gut–brain–microbiota, which is based on a bilateral communication system through signaling from the gut microbiota to the brain and from the brain to the gut microbiota by involving neural, endocrine, immune and metabolic links [20].

Gastrointestinal disturbances are usually related to an impaired functional link between the intestinal microbial ecosystem and macroorganisms, which could cause even colorectal cancer development, the world’s third most common malignant tumor [26]. The gastrointestinal tract’s proper functioning depends on the composition of inhabiting microbiota, where the intestinal alkaline phosphatase (IAP), a membrane-bound glycoprotein secreted into the lumen by enterocytes, is a part of the defense system in the small intestine [27]. When studying the fundamental molecular mechanisms based on delicate interactions between cells and their environment, the preceding in vitro approach is more appropriate. Widespread models investigate these mechanisms based on the human adenocarcinoma cell line HT29, which, depending on culture conditions, can differentiate into polarized monolayers of mucus-secreting absorptive cells [28]. They express similarities with mature enterocytes in the differentiated phenotype because they form a monolayer, a typical apical brush border, and express brush-border-associated hydrolases. For this reason, the HT29 cell line is a valuable model for studying the biology of colon cancers, food digestion and bioavailability [29]. Produced by enterocytes, alkaline phosphatase activity is a marker used for their differentiation [30,31]. Moreover, we could predict the cellular fate in such models by tracking the balance between B-cell lymphoma protein 2 (Bcl-2)-associated X (Bax) and Bcl-2. They act as a promoter and an inhibitor of apoptosis, respectively, and play a crucial role in tumor progression or the inhibition of intrinsic apoptotic pathways [32,33,34].

Based on the factors mentioned above, the current study aimed to determine whether the products obtained from potentially active autochthonous *Lacticaseibacillus paracasei* strains isolated from mountain redwood ants anthills reveal anticancer and probiotic properties and possess potential for application as nutraceutical additives.

## 2. Materials and Methods

### 2.1. Isolation, Culturing and Identification of the Lacticaseibacillus paracasei Strains

The isolation of LAB from eight mountain anthills was performed with wooden sticks placed into them. The anthills, populated by redwood ants (*Formica rufa* L.), were located in Sinite Kamani National Park, Sliven, Bulgaria.

The wooden sticks were placed into sterile containers with skimmed milk for bacteriological purposes. The samples were cooled, transported to the laboratory and then incubated at 37 °C for 24 h. One milliliter of the samples with visual coagulation was transferred in 9 mL of sterile saline (0.85% NaCl, *w*/*v*), supplemented with peptone (0.1%, *w*/*v*; Oxoid, UK), and then serial dilutions from homogenates were prepared. One milliliter aliquots of the 10^−4^, 10^−5^, 10^−6^ and 10^−7^ dilutions were pour-plated in MRS agar (Oxoid, UK) to isolate *Lactobacillus* strains (each sample was plated in duplicate). After incubation at 37 °C for 48 h, the morphology of the cells was observed by light microscopy after Gram staining. The strains were tested for the absence of catalase by directly applying 3% H_2_O_2_ to the colonies. The Gram-positive and catalase-negative rods were streaked three times on MRS agar (Oxoid, UK) to obtain pure cultures.

The bacterial isolates that were defined as *Lactobacillus* spp. on the basis of the preliminary test results were further classified by using the ARDRA technique [6] and species-specific PCR with particular primer sets as follows: *L. paracasei* (5′-CCCACTGCTGCCTCCCGTAGGAGT-3′ and 5′-CACCGAGATTCAACATGG-3′) [35] and *L. rhamnosus* (5′-CAGACTGAAAGTCTGACGG-3′ and 5′-GCGATGCGAATTTCTATTATT-3′) [36].

### 2.2. Cell Line and Culturing

Human colorectal adenocarcinoma cells (HT29, HTB-38™, ATCC, Washington, DC, USA) were towed and refreshed in basal media (BM, containing DMEM, 10% FBS and an antibiotic) in humidified 37 °C and 5% CO_2_ conditions. At 90% confluency, the cellular monolayer was trypsinized, and the cells were seeded on 24-well plates at a concentration of 5 × 10^4^ /mL. Further, the cells were left to settle for twenty-four hours and then treated with supernatants (SNs) derived from four strains of *Lacticaseibacillus paracasei*: P4, C8, C15 and M2.1. The strains were cultured in MRS, and 24 h later, their SNs were centrifuged for ten min at 9000 rpm and filtered through a 20 µm syringe filter, and the pH was measured (Table 1).

The total amount of each SN was divided into two parts, where half was neutralized to pH = 7. Further, the following percent dilutions of each type of SN (native and neutralized for each strain) in BM were prepared: 5%, 10%, 20% and 40%. The preliminary testing showed severe cellular mortality at higher concentrations, which was estimated at 100% in both native and neutralized 80% and 100% SN solutions. Each SN concentration (5%, 10%, 20% and 40 %) was tested on six wells of a 24-well plate of non-confluent HT29 cells, cultured in 0.6 mL experimental media from native and neutralized P4, C8, C15 and M2.1. The HT29 cells were treated for 24 h at 37 °C in humidified 5% CO_2_ conditions.

In addition to the aforementioned main groups, the following controls were established: MRS in native with pH 5.6 and neutralized with pH = 7; Lacto—BM supplemented with lactic acid (pH = 4), and neutralized Lacto with pH = 7 (Osmo) in the following concentrations in BM for each pH, respectively: 5%, 10%, 20% and 40%. These additional treatments aimed to evaluate the pure biological effects of bioactive lactobacilli products on the cellular viability and proliferation of the tumor cells, eliminating the MRS’s influence that constitutes the pH and osmolarity of the culturing media.

The micrographs were taken by an inverted microscope, Leica DM1000 LED (Heerbrugg, Switzerland), equipped with a 5.0 megapixel resolution DMi1 camera and the software platform Leica Application Suite Core.

### 2.3. MTT Assay

The estimation of treatments’ impact was based on a colorimetric quantification method, where the ability of the live cells to reduce 3-(4,5-dimethylthiazol-2-yl)-2,5-diphenyltetrazolium bromide (or MTT) to insoluble purple formazan was measured. In brief, after treatment of the HT29 cell for 24 h, the experimental culturing media was replaced by 5 mg/mL MTT dissolved in BM for 2 h, kept at 37 °C, in humidified 5% CO_2_ conditions. After this, the medium was replaced with the same amount of acidified isopropanol. The samples were incubated for 10 min at room temperature in an orbital shaker to extract intracellular formazan crystals. The optical density (OD) was measured spectrophotometrically using a microplate reader, Synergy™ LX Multi-Mode (BioTech^®^, Santa Clara, CA, USA), at a 570 nm wavelength. The magnitude of absorption was proportional to the number of intracellular formazan crystals eluted by isopropanol. The raw data from the MTT assay were additionally calculated as a percentage of the mean OD value for each sample accordingly, as described by Chen et al., 2017 [37]:Inhibition (%) = [(OD treated − OD controls)/OD controls] × 100%.(1)

Alkaline phosphatase activity experiment: The cells were cultured in the described manner and conditions in BM up to 80% confluence. After this stage, the cells were divided into two experimental groups: in the differentiated (Figure 1A) group, the cells were cultured in differentiating media for 21 days, as described by Gout et al., 2004 [38], and treated for 24 h with experimental media containing 10% SNs from P4, C8, C15 and M2.1; in non-differentiated (Figure 1B) cells, the medium was changed to experimental media having only 10% SNs from P4, C8, C15 and M2.1 for 24 h.

Additionally, the cells from both groups were cultured in 10% MRS for 24 h to follow its effect. After the treatment period, the experimental culturing medium was collected, and the ALP activity was estimated colorimetrically with a BS-120 Chemistry Analyzer (MINDRAY, Guangzhou, China) using ALP reagent (Biolabo reagents, Maizy, France). The ALP activity was also measured in SNs from P4, C8, C15 and M2.1 themselves.

### 2.4. Gene Expression Analysis (RT-qPCR)

According to the manufacturer’s instructions, the tRNA was isolated and purified using a Universal RNA Purification Kit (EURx, Gdansk, Poland). Its quantity and quality were determined spectrophotometrically by absorbance at 260 and 280 nm with the Agilent Cary 60 UV/Vis. The reverse transcription was performed with the RevertAid First Strand cDNA Synthesis Kit (Thermo Scientific, Waltham, MA, USA), and the cDNA was stored at −20 °C. The expression of the target genes, Bax, Bacl-2 and IAP, was evaluated using the SYBR-based real-time RCR thermocycler Gentier 96E (Xi’an Tianlong Science and Technology, Xi’an, China), using SG qPCR Master Mix (EURx, Gdansk, Poland) and gene-specific primers from Sigma Aldrich (Merck KGaA, Darmstadt, Germany) (Table 2).

The amplification was performed by two-step qPCR. The initial denaturation was prolonged for 3 min at 95 °C, followed by 40 cycles (denaturation for three seconds at 95 °C and annealing/extension for 30 s at 60 °C). All samples were analyzed in duplicate. The primers for target and housekeeping genes were designed, and the product length and annealing temperatures were determined by the web-based software Primer Blast NCBI (www.ncbi.nlm.nih.gov, accessed on 1 April–30 May 2022). The row data were normalized to the two housekeeping genes, GAPDH and Actb, whose stability value was determined by NormFinder software [39]. The IAP relative mRNA expression was plotted as a fold change to the reciprocal MRS HG/LG control.

### 2.5. Statistical Analysis

The statistical analysis was performed by Statistica v. 7.0 (StatSoft Inc., 2004, Tulsa, OK, USA). Each experiment was repeated four times (six for the MTT assay) and the measurements were performed in duplicate, where the mean of each value was considered for further statistical analysis. Based on the above, the mean value for each group and the standard error of the mean (±SEM) were evaluated. The treatment effect was estimated by a nonparametric Mann–Whitney U-test, where *p* < 0.05 values were considered significant. The LSD test was used to estimate the significant differences between groups’ mean values in the MTT assay experiment. The influence of control media was evaluated by correlation analysis, where the *p* < 0.05 correlation coefficient values were considered significant.

## 3. Results

### 3.1. Identification of the Lactobacillus spp. Isolates

From the eight collected samples, we annotated 31 bacterial isolates for further analysis. After initial screening, eleven strains were identified as *Lacticaseibacillus paracasei*. Based on the results of the previous study [7], we have chosen four strains with the best antimicrobial properties to analyze their potential anticancer and probiotic activity.

### 3.2. MTT Assay

The MTT assay used to measure the pure impact of the treatments showed that, except those from C8, in both native and neutralized 5% and in neutralized 10% concentrations, the SNs improved the cellular viability in human HT29 cells (Table 1). The magnitude of the observed effects differed within groups. When the pH of the SNs was neutralized, the lower concentrations of P4 significantly increased cellular viability (*p* < 0.05). The MRS inhibited cellular vitality in concentrations above 5% in native and above 10% in neutralized pH since the lactic acid provoked an adverse effect in all tested groups. Significant differences between the SNs’ effects in 5% and 10% treatments were not established in any group.

The additional correlation analysis revealed a significant negative relationship between MRS and both native (r = −0.9986, *p* < 0.01) and neutralized (r = −0.9592, *p* < 0.05) SNs from M2.1 in 5% concentrations. The lactic acid correlated considerably positively with 5% neutralized SNs from C15 (r = 0.9985, *p* < 0.01) and negatively with 10% native SNs from C8 (r = −0.9720, *p* < 0.05). The correlations between native and neutralized 20% and 40% SNs from one side and native and neutralized MRS and Osmo from the other were insignificant.

Based on the above-listed results, the subsequent experiments were performed on proliferating (non-differentiated) and non-proliferating (differentiated) HT29 cells cultured in 10% neutralized supernatants from P4, C8, C15 and M2.1 strains of *Lacticaseibacillus paracasei*, since this was the higher concentration that was safe for the cells, without cytotoxic signs.

### 3.3. Alkaline Phosphatase Experiment

Before the cell treatment, alkaline phosphatase activity was not detected in 10% SN media from P4, C8, C15, M2.1 and MRS. The 10% neutralized SNs from *Lacticaseibacillus paracasei*-supplemented culturing media caused, in differentiated HT29 cells, a significant (*p* < 0.001) rise in ALP activity, with an average of 18% in all treatments, compared to its relative MRS control (Figure 2).

Differentiated cells showed significantly (*p* < 0.001) higher ALP activity than non-differentiated ones in all SN treatments. The most substantial effects were observed in the C8 (20%) and M2.1 (21%) groups, where the ALP activity was significantly higher than in P4. In non-differentiated cells, the SNs from P4 and C8 were detected as a insignificant suppression of ALP activity, since the C15 and M2.1 treatments had an adverse influence. There was a lack of significant effects of the supernatants compared to the relative MRS in the same group.

### 3.4. IAP Gene Expression

The IAP expression was significantly (*p* < 0.05) downregulated in the differentiated M2.1 group and upregulated in the non-differentiated (HG) P4 and M2.1 groups (Figure 3).

There were significant differences between non-differentiated and differentiated IAP mRNA expression in P4, C15 and M2.1 treatments. The intergroup comparison in differentiated cell groups revealed significant downregulation in P4 and M2.1 compared to C8, C15 and M2.1 and C8 and C15, respectively.

### 3.5. Bax/Bcl-2 Quantity Expression Ratio

Compared to the controls, the P4 10% SNs were significantly elevated (*p* < 0.05) in proliferating cancer cells. At the same time, in differentiated groups, P4 and M2.1 SNs significantly reduced (*p* < 0.05) the Bax/Bcl-2 ratio (Figure 4).

Significant differences (*p* < 0.05) were observed between P4 and C8, C15 and M2.1 and between M2.1, C8 and C15 in differentiated cells.

The summarized net effects of SNs from P4, C8, C15 and M2.1 LAB strains are presented in Table 3.

## 4. Discussion

The main achievement in the current study was that extracellular products in SNs isolated from four LAB strains caused opposite changes in the early predictive pro-/anti-apoptotic Bax/Bcl-2 ratio and ALP activity in proliferating and non-proliferating HT29 cell lines. Our findings elucidate the dual anticancer and probiotic potential of isolated strains (Table 3), typical features for probiotics in general and new for the LAB, isolated from anthills populated by redwood ants.

The MTT results revealed that, except C8, all the LAB strains suppressed the proliferation of the HT29 cells in concentrations above 10% for native SNs and above 20% for neutralized SNs (Table 1). The increasing Bax/Bcl-2 ratio in all SN treatments in non-differentiated cells indicated the activation of the early predictive pro-apoptotic marker, even against the backdrop of the slightly active proliferation at the 10% concentration, and revealed the anticancer potential of the abovementioned SNs. Only in the C8 strain were inhibitory growth features constantly expressed in all treatments. Similar results were reported by Karimi Ardestani et al., 2019 [24], where heat-killed *Lactobacillus brevis* and *Lacticaseibacillus paracasei* bacteria inhibited the proliferation of HT29 cells and induced apoptosis. The researchers reported that the observed changes were time-, dose- and strain-dependent, which is supported by our results.

Since we used SN treatments, the observed effects could also be related to the extracellular vesicles released into the MRS culture media by *L. paracasei* during incubation. These vesicles serve as communication molecules between cells, and colorectal cancer cells can intake them. Further, they can inhibit the growth of colorectal cancer cells in vivo and in vitro by inducing apoptosis through the 3-phosphoinositide-dependent protein kinase-1 (PDK1)/AKT/Bcl-2 signaling pathway [26]. Reduced expression of Bcl-2 in HT29 cells is also detected in heat-killed probiotic bacteria, including *L. paracasei*, where growth inhibition ability and apoptosis induction in HT29 cells is observed [24].

The inhibition of Bcl-2 is insufficient to estimate the significance and magnitude and predict the outcome of the cancer suppression abilities associated with the LAB treatments. The relative ratio between pro-apoptotic and anti-apoptotic proteins determined by the division of means for Bax expression to Bcl-2 expression levels (Bax/Bcl-2 ratio) is more informative and has been regarded as a prognostic marker in various cancers [32,33,34]. Similar studies using the Bax/Bcl-2 ratio as a marker in experimental conditions investigating the LAB impact on a porcine intestinal cell line [40] and HT29 [41] are reported. Moreover, in melanoma cells, the Bax/Bcl-2 ratio < 1.00 is considered for resistant cells since the Bax/Bcl-2 ratio is >1.00 for sensitivity [42]. Transferring these findings to our results, all the treatments in non-differentiated cells elevated the Bax/Bcl-2 ratio above 1.00, which indicates again that the current four SNs achieved from LAB strains originating from different anthills induce apoptosis in human colorectal adenocarcinoma HT29 cells (Figure 4). We also established that the Bax/Bcl-2 ratio was lower in differentiated than in non-differentiated cells, confirming the same findings in normal and tumor tissue and suggesting that this ratio could be used as a prognostic or predictive marker for the fate of colorectal cancer cells upon treatment [32]. By suppressing/blocking Bcl-2, the apoptotic process in tumor cells can be restored [43]. Therefore, apoptosis induction in cancer cells is one of the most efficient strategies to treat cancer and identify anticancer compounds [24]. In this sense, the induction of apoptotic events in non-differentiated (cancer) cells is the desired outcome, since its retardation in differentiated cells would benefit and prolong their functionality as enterocytes (Table 3).

In the differentiated group, in contrast to C15, the observed significant ALP activity was related to the downregulation of IAP in P4, C8 and M2.1 (Figure 3). The effect is probably due to the negative regulatory feedback between gene expression and enzyme activity. Since IAP is an enterocyte-differentiating marker as well [30,31], its low expression and the insignificant changes in ALP activity in non-differentiated cells are understandable.

In contrast to the effects in non-differentiated cells, the reduced Bax/Bcl-2 ratio and increased ALP activity in differentiated groups in the P4 and M2.1 treatments revealed anti-apoptotic and pro-differentiation abilities, underlining their probiotic features. IAP is believed to play a vital role in maintaining gut homeostasis and mitigating inflammatory-mediated disorders related to intestinal inflammation, dysbiosis, bacterial translocation and systemic inflammation in animals and humans [44]. It is essential in fat and phosphate metabolism [45], inactivates bacterial pathogens by producing antimicrobial compounds [46] and promotes the bacterial colonization of the intestine with commensal organisms [27,44]. Since antimicrobials are often used in human and veterinary medical practices, emerging probiotics could be an alternative intervention measure to prevent bacterial infections [47].

Conversely, the gut microbiota could exchange IAP gene expression and enzyme activity [27]. IAP’s role in detoxifying lipopolysaccharide and preventing bacterial invasion across the gut mucosal barrier has also been demonstrated [48]. Moreover, IAP overexpression improves intestinal barrier function by maintaining the integrity of the mucin layer and attenuating intestinal lipid absorption, reducing plasma lipids and attenuating the development of Western-type diet-induced atherosclerosis [49]. Recently, Jang et al. [50] reported four novel canine probiotic strains, including *L. paracasei*, and found changes in the clinical parameters of blood and microbial abundance in feces under commercial probiotic feeding conditions. In this sense, the use of LAB could be a fundamental therapeutic approach to fighting the development of metabolic syndrome, type II diabetes, hepatic steatosis, atherosclerosis and heart disease. This functional cross-talk between the gut microbiota and endogenous alkaline phosphatase could govern the development and progression of a pathological event and determine its different outcomes in food digestion disorders and cancer development. Our data support the probiotic potential of the LAB strains used in our study, where an intense elevation in ALP activity in differentiated HT29 cells was demonstrated in all treatments (Figure 2). Even with increased ALP activity, the SNs from C8 and C15 showed insignificant pro-apoptotic effects within the treatment period and would not be an appropriate choice as a probiotic component (Figure 4, Table 3).

Additionally, their features related to the attenuation of intestinal lipid absorption and the consequent prevention of obesity development are being considered further in our ongoing study.

## 5. Conclusions

The LAB SNs from P4, C8 and M2.1 in non-differentiated proliferating HT29 cells revealed different promising anticancer properties, manifested independently or in a combination of their effects by promoting differentiation and apoptosis and suppressing proliferation. Concerning the probiotic activity of the isolated strains, we found that P4 and M2.1 are the best candidates, causing anti-apoptotic and pro-differentiation events in differentiated cells. Based on our results, we would expect a more substantial anticancer effect by combining these three *L. paracasei* strains. In conclusion, the newly isolated autochthonous *Lacticaseibacillus paracasei* strains in the current study possess nutraceutical potential and can be applied as additives in new dairy functional foods, supporting the treatment and prevention of human colorectal cancer.

## Figures and Tables

**Figure 1 foods-12-00185-f001:**
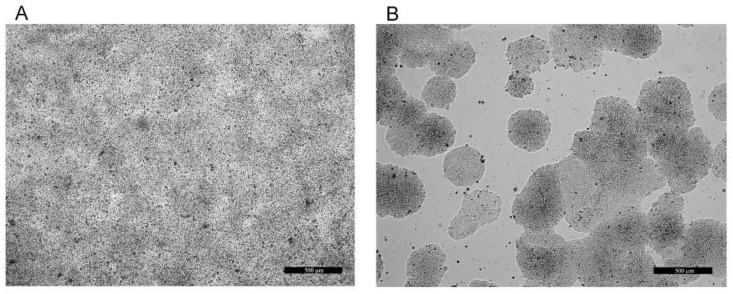
Native micrographs of differentiated (LG, non-proliferating (**A**)) and non-differentiated (HG, proliferating (**B**)) human colorectal adenocarcinoma HT29 cells. Bars = 500 µm.

**Figure 2 foods-12-00185-f002:**
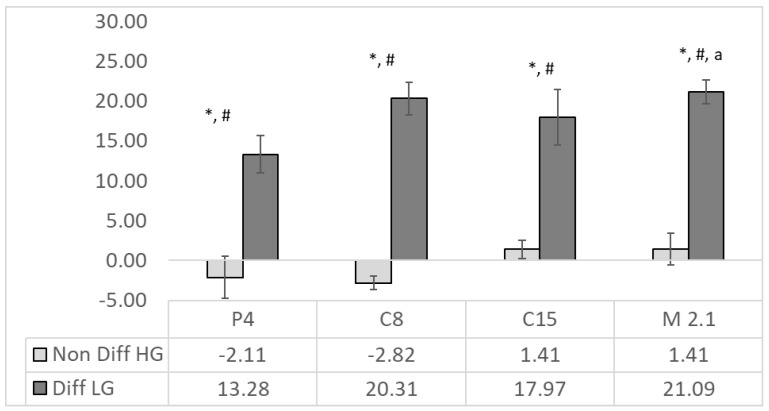
ALP activity (%) in supernatants from C15, M2.1, P4 C8 strains of *Lacticaseibacillus paracasei* in non-differentiated (in light grey) and differentiated (in dark grey) human colorectal adenocarcinoma HT29 cells. The ALP activity is calculated as a % of the MRS (*n* = 4, ±SEM), significance: to the relative MRS control LG/HG for non-differentiated and differentiated cells, respectively—*p* < 0.001 = *; non-differentiated vs. differentiated *p* < 0.001 = #; and differentiated P4 to M2.1 to *p* < 0.05 = a.

**Figure 3 foods-12-00185-f003:**
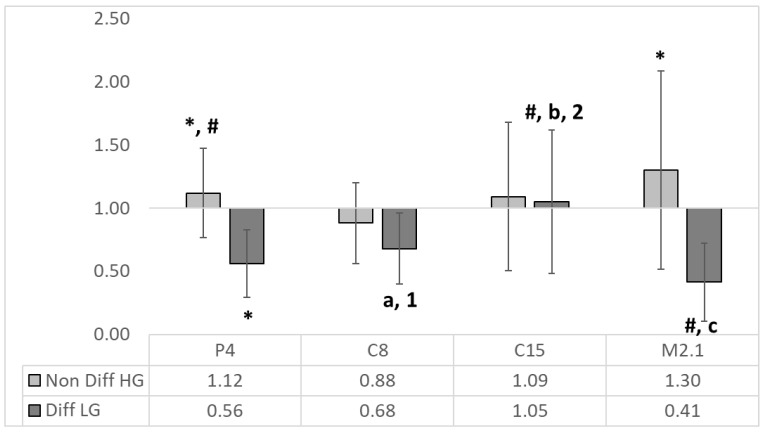
The relative expression of IAP to the MRS HG/LG control (*n* = 4, ±SEM). The established significant differences (*p* < 0.05) are expressed as follows: *—significance to the reciprocal MRS HG/LG control; # significance between groups; the significance in differentiated (LG) group is expressed as follows: with small letters between P4 and C8 (a), C15 (b) and M2.1 (c); with numbers between M2.1 and C8 (1) and C15 (2).

**Figure 4 foods-12-00185-f004:**
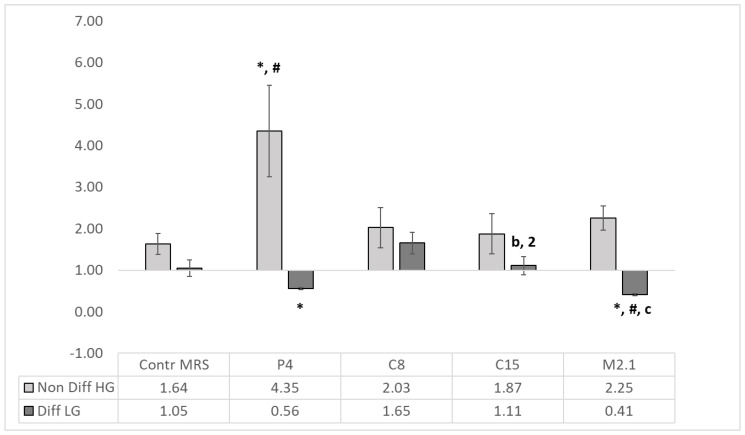
Pro-/anti-apoptotic Bax/Bcl-2 ratio. The absolute qPCR quantities of BAX and Bcl-2 gene expression are presented as Bax/Bcl-2 ratio (*n* = 4, ±SEM). The established significant differences are expressed as follows: * significance to the MRS HG/LG Control; # significance between groups; the significance in differentiated (LG) group is defined as follows: with small letters between P4 and C8 (a), C15 (b) and M2.1 (c); with numbers between M2.1 and C8 (1) and C15 (2).

**Table 1 foods-12-00185-t001:** Hydrogen ion concentration (pH) of MRS after 24 h incubation period of lactobacilli, its neutralization up to pH = 7, and the mean OD (at 570 nm) values estimated by MTT test, representing inhibition/proliferation rate in % of non-confluent HT29 treated for 24 h with different SN concentrations from four strains (P4, C8, C15 and M2.1) of *Lacticaseibacillus paracasei* and the controls, Lacto and Osmo.

	P4 *n* = 6	C8 *n* = 6	C15 *n* = 6	M2.1 *n* = 6	MRS *n* = 6	Lacto *n* = 4
Native pH	pH 3.94	pH 4.40	pH 4.08	pH 3.98	pH 5.6	pH 4.00
5%	3.32 ± 1.57	−5.84 ± 5.09	9.83 ± 3.59*	6.21 ± 3.66	1.44 ± 1.51	−19.42 ± 2.15 ^a^
10%	−3.52 ± 0.64	−13.93 ± 2.11 *	0.51 ± 1.10	9.33 ± 3.37 *	−7.31 ± 3.39 ^a^	−91.03 ± 0.99 *^,a^
20%	−92.97 ± 0.64 *^,a^	−38.81 ± 2.65 *	−83.39 ± 2.13 *^,a^	−90.16 ± 1.73 *^,a^	−20.33 ± 3.19 *	−84.12 ± 2.54 *^,a^
40%	−88.42 ± 0.32 *^,a^	−93.17 ± 0.51 *^,a^	−83.31 ± 1.31 *^,a^	−80.86 ± 0.48 *^,a^	−35.00 ± 1.83 *	−86.99 ± 1.85 *
Neutralized pH	pH 7	pH 7	pH 7	pH 7	pH 7	Osmo_pH7
5%	22.35 ± 3.38 *^,a^	−6.32 ± 3.28	2.30 ± 3.71	9.53 ± 2.39 *	4.34 ± 2.63 *^,a^	−5.51 ± 2.46 *
10%	19.38 ± 2.92 *^,a^	−8.24 ± 2.50	1.15 ± 2.36	8.82 ± 2.34 *^,a^	2.31 ± 1.59	−13.53 ± 4.37 *
20%	0.18 ± 3.51	−37.99 ± 1.59 *	−8.67 ± 1.59 *	−9.20 ± 2.30 *	−12.18 ± 1.48 *	−47.88 ± 1.03 *
40%	−33.97 ± 0.65 *	−49.11 ± 1.04 *	−29.87 ± 2.49	−36.78 ± 2.14 *	−39.24 ± 1.35 *^,a^	−93.78 ± 0.46 *^,a^

In acidic (in orange beckround) and normal (in green backround) pH culturing conditions at 5%, 10%, 20% and 40% concentrations. The inhibition rate is represented as grey background. (*n* = 4–6, ±SEM, to control BM: *p* < 0.05 = *; SNs vs. SNs_pH 7: *p* < 0.05 = a).

**Table 2 foods-12-00185-t002:** qPCR forward and reverse primer sequences and product lengths.

Abbreviation	Full Name	Forward	Reverse	Product Length
BAX NM_001291430.2	BCL-2 associated X, apoptosis regulator (BAX)	CCCGAGAGGTCTTTTTCCGAG	CCAGCCCATGATGGTTCTGAT	155
BCL-2 NM_000657.3	BCL-2 apoptosis regulator (BCL-2)	GCTCTTGAGATCTCCGGTTG	AATGCATAAGGCAACGATCC	186
IAP NM_001631.5	Intestinal alkaline phosphatase	GTATGTGTGGAACCGCACTG	CTGGTAAGCCACACCCTCAT	244
GAPDH NM_002046.7	Glyceraldehyde-3-phosphate dehydrogenase	GAGTCAACGGATTTGGTCGT	TTGATTTTGGAGGGATCTCG	238
Actb NM_001101.5	Actin beta	CGTCTTCCCCTCCATCGT	GGGGTACTTCAGGGTGAGGA	124

**Table 3 foods-12-00185-t003:** Summarized net effects profile of SNs from P4, C8, C15 and M2.1 LAB stains on proliferating (A) and differentiated (B) HT29 cells. The number of arrows represents the magnitude of the impact based on the results presented in Table 1 and Figure 2, Figure 3 and Figure 4. In brackets, we indicate whether the effect is positive or negative regarding the anticancer (in non-differentiated) and probiotic (in differentiated) features of the SN.

A: Non-Differentiated HT29 Cells (Cancer Cells)				
	P4	C8	C15	M2.1
**ALP activity**	↓(−) differentiation	↓(−) differentiation	↑(+) differentiation	↑(+) differentiation
**IAP**	↑(+) differentiation	↓(−) differentiation	↑(+) differentiation	↑↑(+) differentiation
**Bax/Bcl** **Ratio**	↑↑↑(+) pro-apoptotic	↑↑(+) pro-apoptotic	↑(+) pro-apoptotic	↑↑(+) pro-apoptotic
**Net effects**	Anticancer potential	Anticancer potential	Insignificant anticancer potential	Anticancer potential
**B: Differentiated HT29 Cells (Enterocytes)**				
	**P4**	**C8**	**C15**	**M2.1**
**ALP activity**	↑↑(+) differentiation	↑↑(+) differentiation	↑↑(+) differentiation	↑↑(+) differentiation
**IAP**	↓↓(−) differentiation	↓(−) differentiation	↑(+) differentiation	↓↓↓(−) differentiation
**Bax/Bcl Ratio**	↓(+) anti-apoptotic	↑↑(−) pro-apoptotic	↑(−) pro-apoptotic	↓(+) anti-apoptotic
**Net effects**	Probiotic potential	Insignificant probiotic potential	Insignificant probiotic potential	Probiotic potential

## Data Availability

The datasets generated for this study are available on request to the corresponding author.

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
