# Peer review of "Evaluation of the Anticancer and Probiotic Potential of Autochthonous (Wild) *Lacticaseibacillus paracasei* Strains from New Ecological Niches as a Possible Additive for Functional Dairy Foods"

_foods, 2023, doi:10.3390/foods12010185_

Round 1

Reviewer 1 Report

In the paper by Vachkova et al 

seems somewhat premature to define Lacticaseibacillus paracasei as an anticancer and with probiotic properties and to be used as food additives.

The data supporting this hypothesis are poor, uncorrelated and statistically weak.

The only HT-29 cell line chosen as a model for the study, does not offer a realistic view of the effect of L.paracasei. 

The choice of performing a MTT on undifferentiated cells is not clear. How extent can L. paracasei alone influence the MTT on cells not ready to be treated, but placed only in a condition of non-differentiation? this is a not physiological condition. Cells in their real condition are differentiated and able of reaching a physiological condition that is more compliant with the treatments received.

MTT exp must be repeated.

The succession of exp is only a YES\NO of the effect of L.paracasei on the analyzed parameters APL, Bax\Bcl2 ratio. It is evident that in this condition cell differentiation plays a fundamental role, but no mechanism, hypothesis or description is offered for the achievement of the data. Fundamental information on numbers of experiments, treatments and mechanism hypotheses is missing. The effects, alone, on the Bax\Bcl2 ratio are not sufficient to define strains with anti-carcer effect. Many more effects (proliferation, growth inhibition\stimulation, increase\decrease of specific genes involved in cancer proliferation, and above all in vivo effects of the same effects seen in cells) are necessary to substain this hypotesis.

The some effects described in the discussion should be included in the data to make the reader understand the purpose of the work and the results obtained.

A more fine  characterization of the differnt strains used in the paper is crucial.

Finally, to make reading the data more fluid, summary tables are needed, and more data must be included to support the hypotheisis.

Author Response

The authors of the article thank the reviewers for their professional and constructive suggestions for improving the manuscript, as well as for correcting some errors and inaccuracies. In order to clarify the information provided, we have made revisions as reviewers recommended. The revisions were highlighted by using the "Track Changes" function in Microsoft Word, so that changes are easily visible. Our detailed explanations, point by point, to each of the reviewers are as follows:

Point 1: seems somewhat premature to define Lacticaseibacillus paracasei as an anticancer and with probiotic properties and to be used as food additives.

Response 1: It has already been proven that, to some extent, LABs, especially Lactobacilli demonstrated anticancer activity. Since the source of the LABs is unusual, we aimed to evaluate if the extracellular products from chosen strains with demonstrated antimicrobial properties additionally possess anticancer possibilities. Based on this, we changed the title from: “Evaluation of anticancer and probiotic properties of autochthonous (wild) Lacticaseibacillus paracasei strains from new ecological niches as a potential additive for functional dairy foods” to “Evaluation of the anticancer and probiotic potential of autochthonous (wild) Lacticaseibacillus paracasei strains from new ecological niches as a possible additive for functional dairy foods.”

Point 2: The data supporting this hypothesis are poor, uncorrelated, and statistically weak.

Response 2: According to PubMed, more than 300 articles are found regarding the hypothesis of the LAB anticancer activity. Due to the fact that this is only a hypothesis, we have endeavored to confirm or reject it.

The most data obtained in our study are statistically significant. Some of them are at the highest level of significance (p<0.001, 99% confidence level), proving our statements.

Point 3: The only HT-29 cell line chosen as a model for the study does not offer a realistic view of the effect of L.paracasei. 

Response 3: The widely spread protocols annotated the model for such investigations is co-culturing HT29:Caco 2 in different ratios (from 5%:95% to 25%:75%), where after proper treatment, the first cell line mimics functionally Globlet cells, since the second enterocytes (https://pubmed.ncbi.nlm.nih.gov/29787039/). The disadvantage of co-culturing is subsequently separating both cell fractions for further examination at the end of the experiment.

The HT29 cell line can behave as well as tumor and enterocyte cells in proper media conditions, and for that reason, it is considered a pluripotent intestinal cell line. It is widely used and is routinely chosen for such a purpose, especially for testing anticancer substances and cellular survival. During differentiation, they become polarized, forming microvilli and brush-border membranes and express enterocyte-specific enzymes, including alkaline phosphatase. They can successfully differentiate in enterocytes in glucose deprivation conditions (https://pubmed.ncbi.nlm.nih.gov/15350547/)..

Point 4: The choice of performing a MTT on undifferentiated cells is not clear. How extent can L. paracasei alone influence the MTT on cells not ready to be treated, but placed only in a condition of non-differentiation? this is a not physiological condition. Cells in their real condition are differentiated and able of reaching a physiological condition that is more compliant with the treatments received. MTT exp must be repeated.

Response 4: The MTT is a routine assay for estimating cellular viability upon treatment. For adherent cultures, it is assumed to be evaluated on non-confluent cells since some substances could potentiate cellular proliferation, reflecting a higher absorbance rate. We performed the MTT assay on non-confluent cells to investigate mainly the general cytotoxicity of isolated supernatants and to decide which concentration would be born of the cells and would be the most appropriate for the main experiment. Further, that concentration was used for the treatments of both differentiated and non-differentiated cells (https://www.sciencedirect.com/topics/biochemistry-genetics-and-molecular-biology/mtt-assay).

 Point 5: The succession of exp is only a YES\NO of the effect of L.paracasei on the analyzed parameters APL, Bax\Bcl2 ratio. It is evident that in this condition cell differentiation plays a fundamental role, but no mechanism, hypothesis or description is offered for the achievement of the data.

Response 5: Both control groups (differentiated and non-differentiated) were cultured in 10% MRS broth and basal media to eliminate the effect of MRS and the observed changes representing pure effects of the SNs. We have established that the effects are rather strain-specific than differentiation-dependent (Fig. 2 and Fig. 4), which the controls prove. Further, the results are also presented in graphs, not tables, since we did not want to duplicate the same results in two presenting ways, and if the readers need to follow a particular parameter, the value can be achieved from there. In this meaning, the condition that the ratio is less or higher than one only border the pro-apoptotic and anti-apoptotic conditions. Its magnitude could be estimated additionally by the specific value from the graphs.

 Point 5: Fundamental information on numbers of experiments, treatments and mechanism hypotheses is missing.

Response 5: The following sentence was added to section 2.5  Statistical analysis:  “Each experiment was repeated four times (six for the MTT assay) and the measurements were performed in duplicates where the mean of each value was considered for further statistical analysis.”

The number of replicates is shown in the text below each table and graph.

The hypothesis we used is based on recent findings suggesting that the LAB metabolites can inhibit the growth of colorectal cancer cells in vivo and in vitro by inducing apoptosis through the 3-phosphoinositide-dependent protein kinase-1 (PDK1)/AKT/Bcl-2 signaling pathway which could be indirectly presented by Bax\Bcl2 ratio.

 Point 6: The effects, alone, on the Bax\Bcl2 ratio are not sufficient to define strains with anti-carcer effect. Many more effects (proliferation, growth inhibition\stimulation, increase\decrease of specific genes involved in cancer proliferation, and above all in vivo effects of the same effects seen in cells) are necessary to substain this hypotesis.

Response 6: Bax/Bcl2 ratio is proposed as a routine predictive marker in the diagnosis and prognosis of many human tumors (https://pubmed.ncbi.nlm.nih.gov/?term=bax+bcl-2+ratio). It is the most significant and informative marker for estimating tumor sensitivity to cytostatics and radiation therapy. 

As mentioned above, the MTT assay is done on non-confluent cells to trace if the supernatants in different concentrations provoke cellular cytotoxicity or proliferation. We agree that many more effects could be traced, but the primary purpose of the current study was to evaluate the potential of the isolated strains for further studies. Such experiments are currently running and will be the subject of the additional report, including in vivo studies.

Point 7: A more fine characterization of the differnt strains used in the paper is crucial.

Response 7: We agree that additional investigations are necessary for fully revealing and characterizing the isolated strains' features; this is a central topic of other ongoing research. However, in the current study, we did not utilize live bacterial cells in the experiments. Instead, we investigated the probiotic and anticancer potential of supernatants from the chosen strains with good antimicrobial activity following the recently published data suggesting that the activity of LAB metabolites is not limited only to the antibacterial effect but also includes immunomodulatory effect and cancer control.

Point 8: Finally, to make reading the data more fluid, summary tables are needed, and more data must be included to support the hypothesis.

Response 8: Please find the summary table in the corrected version.

Reviewer 2 Report

This article titled “Evaluation of anticancer and probiotic properties of autochthonous (wild) Lacticaseibacillus paracasei strains from new ecological niches as a potential additive for functional dairy foods” evaluated the potential anti-cancer ability of Lactobacillus paracasei strains which isolated from mountain anthills at the cellular level. This study is interesting, especially in terms of the source of the strains. And as a series of cell tests in vitro, this research has been enough to reach the point of view. However, I have some questions and suggestions that need to be explained by the authors.

1. Although sufficient background information has been included in the introduction, I think the research progress of probiotics in vitro evaluation experiment at the cellular level should be mentioned.

2. The authors identified strains in previous study and evaluated their antimicrobial properties (Ref. 7), however, have these strains been tested for intestinal tolerance (such as acid resistant and bile salt resistant)? After all, one of the prerequisites for a strain to be called a "probiotic" is that it can reach the intestinal tract alive.

3. Some formatting or spelling errors should be corrected. For example, does the authors mean 5×104 cell/mL (Line 120)?.

Author Response

The authors of the article thank the reviewers for their professional and constructive suggestions for improving the manuscript, as well as for correcting some errors and inaccuracies. In order to clarify the information provided, we have made revisions as reviewers recommended. The revisions were highlighted by using the "Track Changes" function in Microsoft Word, so that changes are easily visible. Our detailed explanations, point by point, to each of the reviewers are as follows: 

Point 1: Although sufficient background information has been included in the introduction, I think the research progress of probiotics in vitro evaluation experiment at the cellular level should be mentioned.

Response 1: Line 129-131: We add the following sentence: When studying the fundamental molecular mechanisms based on delicate interactions between cells and their environment, the preceding in vitro approach is more appropriate.

 Point 2: The authors identified strains in previous study and evaluated their antimicrobial properties (Ref. 7), however, have these strains been tested for intestinal tolerance (such as acid resistant and bile salt resistant)? After all, one of the prerequisites for a strain to be called a "probiotic" is that it can reach the intestinal tract alive.

Response 2: We agree with the reviewer that resistance to degradation by gastric juice and bile salts is one of the most important selection criteria for the probiotic LAB if we talk about live cells. However, in the current study, we investigated the probiotic potential of active metabolites from the chosen strains with a good antimicrobial activity following the recent findings suggests that the activity of LAB metabolites is not limited only to the antibacterial effect but also includes immunomodulatory effect and cancer control. On the other hand, the LAB and their metabolites can be lyophilized, included in trace materials, encapsulated, or coated in various farmaceutical forms. There are many ways to protect and enable them to reach the intestines. To what extent the LABs and their products could be preserved and resist the gastric acid environment unchanged could be a matter of further research.

Point 3: Some formatting or spelling errors should be corrected. For example, does the authors mean 5×104 cell/mL (Line 120)?.

Response 3: The errors are corrected.

Reviewer 3 Report

The author evaluated anticancer and probiotic properties of autochtho- 2 nous (wild) Lacticaseibacillus paracasei strains from new ecological niches as a potential additive for functional dairy foods.
The topic is of great importance particularly for the nutraceutical based industries & scientists working on probiotics based functional foods.
It adds to anticancer and probiotic properties of autochtho- 2 nous (wild) Lacticaseibacillus paracasei strains from new ecological niches. The author has discussed & studied all the required parameters.
The conclusions are consistent with the evidence and arguments presented.
The references are also up to date.

Author Response

The authors of the article thank the reviewers for their professional and constructive suggestions for improving the manuscript, as well as for correcting some errors and inaccuracies. In order to clarify the information provided, we have made revisions as reviewers recommended. The revisions were highlighted by using the "Track Changes" function in Microsoft Word, so that changes are easily visible. Our detailed explanations, point by point, to each of the reviewers are as follows:

The reviewer there is no comments and suggestions for authors. We thank for it positive assessment of the article.

Round 2

Reviewer 1 Report

The authors have resolved all the submitted concerns quite well.

The decision to modify the title was much appreciated.

The choice of the MTT remains not shared by the reviewer, but can be overcome by making the effort to respond to the other requests.

The addition of the tables, although they duplicate the data already present in the graphs, allows for a much more precise comparison view.

The inclusion of references adds value to the formulated hypotheses, but a more precise reference on the Bax\Bcl2 ratio as a marker should be inserted, in experimental conditions such as those proposed by the authors, i.e. in the presence of LA, to give strength to the proposed concept.

Finally, it would be appropriate to reinforce the concept of "potential" effect of LA’s also in discussion.

After these small changes, the paper can be considered accepted by this reviewer.

Author Response

The authors of the article thank the reviewers for their professional and constructive suggestions for improving the manuscript, as well as for correcting some errors and inaccuracies. In order to clarify the information provided, we have made revisions as reviewers recommended. The revisions were highlighted by using the "Track Changes" function in Microsoft Word, so that changes are easily visible. Our detailed explanations, point by point, to each of the reviewers are as follows:

Reviewer 1: Comments and suggestions for authors

 Point 1: The choice of the MTT remains not shared by the reviewer, but can be overcome by making the effort to respond to the other requests.

Response 1: The MTT assay was used as preliminary screening to evaluate the influence of additional factors, such as pH and osmolarity, which could also affect cellular vitality. MTT is the usual choice for toxicological studies and according to the manufacturer:”The HT29 cell line … has applications in cancer and toxicology research”. In that case, to evaluate the correct percent concentration for further experiments, we choose to perform the MTT on the SN-treated undifferentiated HT29 cell line as it is suitable to test the primary toxicity of the supernatants. For example, if we would test a 20% concentration of the supernatants, which showed up to 92% cytotoxic in undifferentiated cells, we wouldn’t be able to follow the probiotic properties of the substances in differentiated cells, since they would be dead.

Point 2: The addition of the tables, although they duplicate the data already present in the graphs, allows for a much more precise comparison view.

Response 2: Please find the data tables to the graph 2, 3 and 4.

Point 3: The inclusion of references adds value to the formulated hypotheses, but a more precise reference on the Bax\Bcl2 ratio as a marker should be inserted, in experimental conditions such as those proposed by the authors, i.e. in the presence of LA, to give strength to the proposed concept.

Response 3: Additional text with references is included in Page 10, line 331-333.

Point 4: Finally, it would be appropriate to reinforce the concept of "potential" effect of LA’s also in discussion.

Response 4: We have made corrections in the text to reinforce the "potential" effect of LABs.